# Extracellular Vesicle-microRNAs as Diagnostic Biomarkers in Preterm Neonates

**DOI:** 10.3390/ijms24032622

**Published:** 2023-01-30

**Authors:** Emily A. Schiller, Koral Cohen, Xinhua Lin, Rania El-Khawam, Nazeeh Hanna

**Affiliations:** 1Department of Foundational Medicine, New York University Long Island School of Medicine, Mineola, NY 11501, USA; 2Department of Pediatrics, Division of Neonatology, New York University Langone Long Island Hospital, Mineola, NY 11501, USA

**Keywords:** neonate, prematurity, bronchopulmonary dysplasia, necrotizing enterocolitis, hypoxic-ischemic brain damage, extracellular vesicles, miRNA, biomarkers

## Abstract

Neonates born prematurely (<37 weeks of gestation) are at a significantly increased risk of developing inflammatory conditions associated with high mortality rates, including necrotizing enterocolitis, bronchopulmonary dysplasia, and hypoxic-ischemic brain damage. Recently, research has focused on characterizing the content of extracellular vesicles (EVs), particularly microRNAs (miRNAs), for diagnostic use. Here, we describe the most recent work on EVs-miRNAs biomarkers discovery for conditions that commonly affect premature neonates.

## 1. Introduction

Approximately half a million preterm infants are born annually in the USA. Although improvements in medical care have saved many preterm infants who otherwise may not have survived, they have also resulted in more infants with increased morbidities, including necrotizing enterocolitis (NEC), bronchopulmonary dysplasia (BPD), and hypoxic-ischemic brain damage (HIBD), poor neurodevelopmental outcomes, retinopathy of prematurity, and other significant morbidities [1,2,3,4]. Furthermore, the societal cost of preterm births in the USA is at least 25 billion USD [4], which does not include caregiver financial costs and emotional stress [4,5]. This has led to a growing emphasis on the quality of life of preterm infants and the early detection and treatment of such morbidities. However, the field of neonatology lacks reliable diagnostic and prognostic tools for many disorders. Recent research has focused on extracellular vesicles (EVs) found in bodily fluids to understand the pathophysiology of pathological conditions and to identify new diagnostic tests to predict and prevent adverse outcomes through early intervention [6,7]. Although EVs have been characterized in bodily fluids from adults, few studies have identified EVs in neonatal biofluids. In this review, we discuss the current literature describing the role of EVs as diagnostic biomarkers and their potential roles in the pathophysiology of some critical illnesses that affect preterm neonates.

## 2. Definition of an Effective Diagnostic Biomarker

A biomarker is a measurable characteristic that indicates normal physiology, pathological processes, or response to exposure or treatment. The FDA-NIH BEST (Biomarkers, EndpointS, and other Tools) [8] categorizes biomarkers based on their application, including: (1) diagnostic biomarkers to detect the presence of a disease or disease subtype, (2) monitoring biomarkers to assess a parameter over time, (3) predictive biomarkers that identify individuals who are more likely to experience a defined outcome after a specific exposure, (4) prognostic biomarkers that indicate the likelihood of a future clinical event, (5) response biomarkers to show a biological response to exposure, and (6) safety biomarkers that are measured before or after exposure to determine toxicity. Understanding these definitions is imperative for identifying clinically useful biomarkers.

Ideal biomarkers should have the following characteristics [8,9,10]: (1) present in peripheral tissues or fluids that are suitable for sample collection from the target patient population and are involved in the pathophysiological process of the disease; (2) present at a sufficiently high concentration to be detected within a reasonable, defined amount of sample; (3) measurable quickly and affordably with robust analytic performance across various clinical settings; and (4) highly sensitive and specific for the disease in the target population and able to differentiate between diseases that might have similar clinical presentations.

In neonates, there is an urgent need for biomarker discovery to inform and enable early decision-making and personalized treatment plans. Previous approaches aimed at the identification of such biomarkers in neonates have been largely limited by several factors, including: (1) attempting to predict a multi-factorial disease that has diverse pathophysiology by focusing on biomarkers involved in only one particular pathway; (2) the difficulty of identifying a noninvasive sampling site that can accurately mirror biological processes occurring in a specific organ; (3) trying to identify biomarkers that distinguish disease processes that are too advanced in the disease course, limiting effective intervention early in the disease process; (4) using non-sensitive detection techniques, or the use of an intricate assay used only in a research lab that cannot be transferred to a clinically applicable assay; and (5) lack of validation of biomarker expression in larger patient cohorts [7,11,12]. These issues are compounded by the delicate clinical status and small blood volume of neonates, which preclude frequent blood draws for biomarker assessment. We believe that EVs obtained from different non-invasive biofluids can be exploited as accurate biomarkers representative of distinct pathological pathways identified early in the disease process.

## 3. EVs as Effective Diagnostic Biomarkers

EVs, which are released by all cells and are ubiquitous in all bodily fluids, are recognized as highly efficient and biologically significant intercellular communication systems [13,14]. EVs are membrane-bound vesicles secreted by cells to mediate cell signaling and deliver cellular contents to target cells. The targeting and uptake of EVs can be specific or non-specific, depending on their protein and lipid composition [15]. There are two major subtypes of EVs: (1) exosomes (50–150 nm in diameter), which form through the fusion of intraluminal vesicle-containing multivesicular bodies with the plasma membrane [16,17] and (2) microvesicles (50–500 nm in diameter), which form through outward blebbing of the plasma membrane [16,17]. Because exosomes originate from the endocytic compartment, their molecular content mainly reflects that of the parental cell [16]; they serve as surrogates of their cells of origin and are recognized as “liquid biopsies” [18,19]. EVs contain various metabolites, nucleic acids, and proteins that alter cell signaling, protein regulation, and gene expression in target cells [20,21].

In this review, we focus on the diagnostic potential of EV-microRNAs (miRNAs). miRNAs are non-coding RNAs that cause the degradation of specific messenger RNAs and post-transcriptional silencing of gene expression in the target cell [15]. While there are many sources of miRNAs, EV-derived miRNAs continue to be the source of choice for circulating miRNAs owing to the quantity, quality, and stability of EV-miRNAs [22]. Furthermore, the isolation of tissue-specific EV-miRNAs may contribute to increased sensitivity of circulating miRNAs as diagnostic biomarkers [23].

EVs are found in various biological fluids, including peripheral blood, umbilical cord blood, saliva, urine, tears, tracheal fluid, and breast milk [21,24,25,26,27]. In addition, EVs can be purified and enriched from these biological fluids to detect EVs and miRNAs that were previously too small in quantity to be identified. This provides researchers with an optimal opportunity to study the EV content associated with various disease processes. EV-miRNAs serve as candidate biomarkers for many diseases, with most studies focusing on their role in cancer diagnosis [19,28]. Increasing reports show that the sorting of miRNAs is an active process. As such, EV-miRNAs reflect the status of the cells from which they are secreted, and a diseased state can be revealed by sampling biological fluids instead of performing a biopsy on pathologic tissues [29,30,31,32,33,34]. Furthermore, EVs derived from pathological tissues may express different surface markers, enabling the specific isolation of such EVs [29].

### Methods of EV Characterization and miRNA Extraction

To develop EV biomarkers, characterization of EVs from target biofluid and quantitative and qualitative analysis of the EV cargos are essential. The concentration, size, and surface zeta potential can be assayed by nanoparticle tracking analysis (NTA), such as Nanosight and Zetaview [35,36]. To further investigate the protein cargos, EVs can be analyzed by single particle interferometric reflectance imaging sensor (SP-IRIS) using the Exo-View platform [36,37]. To characterize EV morphology, electron microscopy (EM) is commonly used. EM analysis can observe the lipid bilayer and differentiate EVs from dense particles such as lipoproteins [38].

More advanced analytical methods have been developed to study morphology in more detail. Hardij et al. introduced atomic force microscopy as an alternative method for visualizing EVs [39]. Using this technique, it is possible to visualize a single EV and the specific surface antigens. Raman microspectroscopy has also been described as an alternative for label-free visualization of EVs [40]. Using a detection platform that combines a microfluidic device and surface-enhanced Raman spectroscopy (SERS), Wang et al. were able to profile four protein biomarkers in serum EVs [40]. Recently, holotomography imaging has been introduced to gain new insights into EV characterization with an optical, contact-free, label-free examination [41]. Conventional protein analysis techniques such as western blots and ELISA can be used to determine the EV fraction’s protein cargo level.

To quantify miRNAs in biological fluids and EVs, total RNA or RNA with small RNA enrichment extraction is performed with RNA extraction kits, such as column-based extraction [42], chloroform–phenol-based extraction [43], magnetic bead extraction [43], then microarray [44], Northern blotting [45], and quantitative reverse-transcription polymerase chain reaction (qRT-PCR) analysis [46]. Among these methods, qRT-PCR is widely preferred over other detection methods because of its high sensitivity and specificity for detecting low levels of circulating miRNAs in plasma and serum. In the qRT-PCR method, cDNA from specific miRNA is reverse transcribed using specific stem-loop RT primers [47,48] or with the poly-A adopter approach [49,50], followed by PCR with specific PCR primers. Since qRT-PCR is a standard technique already employed in research and central clinical laboratories, it is usually the method of choice in the initial discovery and assay development stages. However, the requirement of reverse transcription and indirect measurement renders the qRT-PCR methods limited in robustness and accuracy. In addition, it is extremely complicated, time-consuming, and laborious; as such, it is unsuitable for clinical practice, particularly in a point-of-care setting. Optical fluorescence-based biosensors that detect the hybridization between the miRNAs and their respective complementary mRNA probes are highly sensitive using fluorescence spectroscopy [51,52]. The label-free detection of biomolecules has been a long-standing goal in developing optical biosensors [53,54,55]. The working principle of the biosensor is measuring the change in the intrinsic physical parameter of the biosensor caused by the binding of miRNA molecules. Therefore, the biosensor methods can assay the target miRNA in its natural state, unmodified. This results in a cost-effective, more reliable, easy, and faster real-time biorecognition interaction detection. Another advantage of the biosensor platform is the ultra-small detection volume requirement and extremely low detection limit (down to the attomole level in some cases). A more detailed discussion of the biosensor platform in miRNA detection is beyond the scope of this paper; readers are recommended to consult reviews by Zhang et al. [56], Dave et al. [57], Cacheux et al. [58], and Lai and Slaughter [59].

## 4. EV-miRNAs in Necrotizing Enterocolitis

NEC is an inflammatory intestinal disease characterized by the invasion of gas-forming microbes in the large and small bowel and can cause ischemic necrosis. NEC is one of the most severe neonatal complications, with a 23% mortality rate [3].

The pathogenesis of NEC is complex, and its pathophysiology has not been fully elucidated. In short, excessive activation of epithelial cell toll-like receptor 4 (TLR4) by the ligand lipopolysaccharide in gram-negative bacteria is a crucial element in the inflammatory response associated with NEC [60]. More specifically, TLR4 activation subsequently activates nuclear factor-kappa B (NF-κB), leading to the release of pro-inflammatory mediators [61].

Early detection and management of NEC are critical to infant mortality; however, reliable biomarkers have not yet been discovered [62,63]. This has led researchers to focus on utilizing EVs to understand the pathophysiology of NEC and potentially detect the disease early in its onset. A recent study confirmed the presence of EVs in the urine of premature neonates (<34 weeks gestation) with significantly altered miRNA profiles in neonates with NEC vs. healthy age-matched controls. The profiles included differential expression of miRNA-5703, miRNA-604, miRNA-5186, and miRNA-139-3p (*p <* 0.05, Table 1) [64]. Molecular network analysis revealed that TNF-α, TGF-β, TP53 (which downregulates NF-κB), and RPS19 (which stabilizes TP53) were associated with these differentially expressed miRNAs [64]. Interestingly, miRNA139-3p induces TP53 in cancer models [65]. In an NEC-induced rat model, TP53 and RPS19 expression were downregulated compared with healthy controls [64]. The finding that EV-miRNAs are differentially expressed in the urine of neonates with NEC indicates that EVs have promising potential to not only further the understanding of NEC pathophysiology but also to serve as biomarkers for NEC.

## 5. EV-miRNAs in Bronchopulmonary Dysplasia

BPD is a chronic lung disease characterized by disruption and inflammation in the pulmonary airways and vasculature, with dysregulated repair mechanisms leading to alveolar simplification, fibrosis, and pulmonary hypertension [75]. Premature neonates are born with underdeveloped lungs and inadequate surface area and surfactant production for gas exchange. Therefore, preterm infants are more likely to require ventilation support, predisposing them to ventilator-induced lung inflammation [76].

It was observed that on day 28 of life in premature neonates born at ≤32 weeks gestation, serum EV-miRNA-21 was upregulated in those with vs. without chronic lung disease (*p* = 0.001, AUC = 0.850, Table 1) while no difference in EV-miRNA-21 levels was found at birth [66]. In agreement with this, EV-miRNA-21 was upregulated in hyperoxia-induced neonatal mice serum (*p <* 0.01, Table 1) [66]. EV-miRNA-21 has been implicated in adult lung diseases, including lung adenocarcinoma and ischemic injury, potentially acting through anti-apoptotic effects [77,78]. It is not surprising that EV-miRNA-21 may also serve as a diagnostic biomarker for lung disease in neonates; however, the role of EV-miRNA-21 in BPD pathophysiology remains unclear.

Comparison of EVs from neonates with and without BPD may reveal their role in the pathogenesis of the disease, as well as potential diagnostic biomarkers. When treated with EVs derived from human tracheal aspirate (hTA-EV), neonatal mice experienced alveolar hypoplasia, increased airway resistance, and right ventricular hypertrophy [79]. When only the CD66+ fraction of BPD hTA-EVs was administered, the same degree of alveolar hypoplasia was observed, indicating the involvement of activated neutrophils in the pathogenesis of BPD [79]. Interestingly, a high neutrophil-to-lymphocyte ratio in peripheral blood samples is an early predictor of BPD in preterm infants [80]. Moreover, compared to the non-BPD group, EVs isolated from human umbilical cord venous blood (hUC-VB-EVs) of neonates who later developed BPD showed significantly reduced cell proliferation and capillary tube formation and a greater reduction in endothelial migration in cultured human umbilical vein endothelial cells [67]. miRNA analysis revealed differential expression of miRNA-103a-3p, miRNA-17-5p, miRNA-185-5p, miRNA-200a-3p, miRNA-20b-5p, and miRNA-765 between BPD and non-BPD hUC-VB-EVs (*p <* 0.05, Table 1) [67]. More specifically, in BPD hUC-VB-EVs, miRNA-103a-3p and miRNA-185-5p showed the most significant reduction, whereas miRNA-200a-3p exhibited increased expression [67]. Additionally, the number of hTA-EVs was elevated in premature neonates with severe BPD at 36 weeks post-menstrual age, with most of the hTA-EVs derived from epithelial cells [68]. The number of EVs also increased in BPD-induced mouse bronchoalveolar lavage fluid (BALF) and in vitro in hyperoxia-induced normal human bronchial epithelial cell culture supernatant [68].

There was a decrease in EV-miRNA-876-3p in the hTA of neonates with severe BPD (*p* < 0.05, AUC = 0.917, Table 1) and BPD-induced mouse BALF (*p* < 0.05), which was associated with increased levels of miRNA-876-3p targets, including anti-apoptotic proteins myeloid leukemia 1 and retinoblastoma-binding protein 6 [68]. The treatment of hyperoxia-induced mice with a miRNA-876-3p mimic led to decreased alveolar hypoplasia and neutrophil inflammation [68]. These findings suggest that suppression of miRNA-876-3p contributes to the pathogenesis of BPD, whereas increased expression may reverse lung damage and inflammation. miRNA-876-3p can act as a diagnostic biomarker as well as a therapeutic target for BPD.

Additionally, miRNA-425 and phosphorylated PI3K/AKT are downregulated in the hyperoxic murine lung (*p* < 0.01, Table 1), whereas PTEN is upregulated. Treatment with murine bone marrow mesenchymal stem cell (mBoM-MSC) EVs attenuated these changes [69]. Inhibition of miRNA-425 in mBoM-MSC-EVs increased apoptosis-induced Bax levels and reduced BCL2 expression in cell culture [69]. miRNA-425 may therefore play a preventative and/or protective role in BPD by inhibiting PTEN, leading to the activation of PI3K/AKT [69]. However, it is currently unknown whether there is under-expression of miRNA-425 in the lung tissues of human neonates with BPD, and if so, whether the lack of EV-miRNA-425 can be measured and utilized as a diagnostic or predictive biomarker for BPD.

## 6. EV-miRNAs in Hypoxic-Ischemic Brain Damage

HIBD, which occurs when cerebral perfusion in the brain is disrupted, is clinically well-defined in neonates > 36 weeks of gestation as neurological dysfunction in the setting of low Apgar scores and metabolic acidosis [81]. However, there are challenges in identifying HIBD in premature neonates, as signs of neurological dysfunction may be present at birth due to prematurity itself rather than HIBD [82,83]. There is limited research on the use of EV-miRNAs for the diagnosis of neonatal HIBD. Most studies currently focus on the role of miRNAs in both the pathogenesis and treatment of HIBD [84], with only a subset of these studies focusing specifically on EV-miRNAs. Furthermore, the current work utilizes full-term models; thus, validation in the premature population is still required. Nevertheless, reviewing work conducted on full-term models may inform future research in the premature population and therefore is discussed here.

HIBD alters the expression of miRNA-182-5p (*p* < 0.05, Table 1) and miRNA-342-3p (*p* < 0.05, Table 1) in mouse brains [70], and miRNA-92b-3p is upregulated (*p* = 0.017, Table 1) and miRNA-342-3p is downregulated (*p* = 0.0006, Table 1) in hUC blood of full-term neonates with moderate-to-severe HIBD [71]. However, while these studies did not identify these miRNAs as EV-derived miRNAs, work on HIBD mouse models suggests their presence in EVs, as miRNA-342-3p, miRNA-92b-3p, and miRNA-182-5p are present in hypoxia-preconditioned mouse brain-EVs, which protect against apoptosis in hypoxic-ischemic-induced mice [72]. Additionally, serum miRNA-410, although not specifically isolated from EVs, is a potential biomarker for hypoxic-ischemic encephalopathy because it is significantly decreased in full-term neonates with HIBD (*p* < 0.01, Table 1) [73]. Interestingly, treating cultured mouse neurons with hUC-MSC-EVs reversed hypoxia-induced damage and upregulated miRNA-410 expression [85]. Overexpression of histone deacetylase-1 (HDAC1), a miRNA-410 target, reversed the neuroprotective effects of hUC-MSC-EVs in cultured neurons [85]. Thus, this led to the conclusion that miRNA-410 confers neuroprotection by inhibiting HDAC1 expression [85].

Analysis of brain tissue in HIBD rats showed that miRNA-124-3p levels were significantly decreased (*p* < 0.05, Table 1), and TRAF6 levels were increased (*p* < 0.05, Table 1) compared to those in the control group [74]. Intracerebroventricular injection of mouse bone marrow MSC-EVs (mBoM-MSC-EVs) transfected with a miRNA-124-3p mimic led to higher neurologic assessment scores and less apoptosis and pathological changes in the brain tissue of HIBD rats on histological examination than that observed following mBoM-MSC-EV treatment alone [74]. Furthermore, silencing TRAF6 attenuated hypoxic-ischemic-induced neural damage, whereas upregulation of TRAF6 antagonized the neuroprotective effects of miRNA-124-3p [74]. Overall, there was a downregulation of miRNA-124-3p levels with upregulation of TRAF6 in neonatal rats with HIBD; however, this miRNA was not specifically derived from EVs [74].

## 7. Next Steps in EV-miRNAs Biomarker Development in Premature Infants

EV-miRNAs have the potential to serve as diagnostic biomarkers for conditions affecting premature neonates, such as NEC, BPD, and HIBD. However, research in this area is preliminary, and many challenges must be addressed before this work can be translated into clinical practice. The major challenges are discussed below and illustrated in Figure 1.

### 7.1. Determine the Best Sample Source for Discovering Diagnostic Biomarkers from Neonatal EVs

The source of the biomarkers is crucial when considering how the research presented here will translate into clinical medicine. EVs can be isolated from many sources, such as blood (serum or plasma), urine, saliva, and feces. There are various methods of isolating EVs, which include ultracentrifugation, precipitation, and size exclusion chromatography (SEC) [86,87]. Long processing time, lack of specificity and sensitivity, and high cost are among some of the limitations of these methods [88]. While EVs can serve as diagnostic biomarkers, there are some limitations that may hinder their use, which may be due to the way they are isolated.

Due to their heterogeneous nature, EVs can be very difficult to quantify [89]. Quantification methods include NTA, dynamic light scattering, and tunable resistive pulse sensing [90]. However, these methods have limited use due to their inability to distinguish between lipoproteins and particles of protein aggregates from EVs [91]. Plasma, which has an abundant source of lipoproteins and aggregates of protein, is a biofluid that requires different techniques to quantify and isolate EVs [92]. One feature of EVs that may aid in their quantification and isolation is the presence of transmembrane proteins. These transmembrane proteins may act as EV markers, making them useful during isolation and quantification.

One studied method of isolating EVs is an insulator-based dielectrophoretic device that is capable of isolating EVs from small sample volumes with a short processing time [88]. Another studied method of isolating EVs is advanced mass spectrometry (MS), which is able to distinguish the protein content of EVs under various physiologic and pathologic conditions [93]. Since EVs can reflect the content of their cells of origin, this serves to be beneficial when deciding which biofluid to analyze in specific neonatal disease processes. One of the major hindrances to characterizing EVs in different biofluids, such as urine and blood, is the presence of a higher magnitude of proteins when compared to EVs [94]. As a result, additional isolation methods are used prior to MS to better extract and characterize EVs. Some of these isolation methods focus on the physical property, such as size and density, as well as EVs’ chemical properties, to better isolate EVs [94].

The characterization of EVs from different biofluids can be affected by many factors, including improper storage and processing conditions. One of the major biofluids used to study appropriate storage, collection, and processing conditions is blood, or, more specifically, plasma. The use of anticoagulants when using blood as the biofluid for EV analysis is controversial. Heparin-based anticoagulants are discouraged. Heparin is associated with false-negative PCR readings since heparin competes with enzymes needed for binding to nucleic acid and can bind to EVs as well as block their uptake [86]. Another factor that can play a role in the optimal isolation of EVs is the fasting state of the patient. Some will analyze the blood samples fairly quickly within one hour of collection, while others will collect blood samples after a 12-hour fasting period [86,94]. This is believed to be required for accurate EV acquirement. The storage of biofluid samples is also controversial. Many have stored samples at 4 °C for up to 5 days without any effect on the number of EVs isolated [95]. However, for long-term storage, samples should be frozen at or below −80 °C and repeated freeze-thaw cycles must be avoided to maintain sample integrity [86].

When choosing which biofluid to analyze in neonates, the ease of collection also comes into play. In the neonatal population, due to their low blood volumes and their susceptibility to becoming anemic with even the smallest of blood draws, blood may not be the ideal biofluid to use when trying to identify biomarkers of diseases. As a result, less invasive biofluid samples, such as urine, saliva, or feces, should become more in favor when analyzing and isolating EVs for potential use as a biomarker. Research on EV isolation in feces is limited. A recent study aimed to address this gap in knowledge through the comparison of EV-isolation techniques in healthy adults [95]. In this study, Tris-EDTA-based preservative buffer was added to stool samples, and the samples were centrifuged and vortexed prior to storage at −80 °C. For EV isolation, ultracentrifugation, precipitation, SEC, and ultrafiltration were compared. It was observed that SEC was the method of choice when considering recovery, reproducibility, and purity. Regarding the neonatal population, there are potential miRNA biomarkers for NEC in neonatal fecal samples [96], and EVs have recently been isolated from the first-pass meconium [97]. However, whether EVs are specifically present in preterm neonatal feces and whether EV-specific miRNAs can serve as biomarkers for NEC or other neonatal diseases remain unknown.

Furthermore, EVs have been characterized in neonatal urine [64]. Numerous studies have been published in adults detailing collection and storage protocols to maximize the stability and recovery of urinary EVs, though there is not one standard protocol that has been established. A recent review article analyzing methods for urine EV isolation concluded that once the urine is collected, it should be stored between 0–4 °C and processed within 8 h [98]. During processing, samples undergo centrifugation to remove cells, cellular debris, and urinary protein uromodulin [98]. Urine samples are then stored at −80 °C, a temperature at which EV-miRNAs are stable even after long-term storage [98,99]. There is conflicting evidence on whether protease inhibitors should be added to samples prior to freezing to prevent urinary EV degradation [99]. However, the use of protease inhibitors would substantially increase the cost of urine biomarker discovery [98]. In Galley et al., urine was collected from preterm neonates by placing cotton balls in their diapers [64] and the urine samples were frozen at −80 °C without processing or the addition of a protease inhibitor. Samples were thawed prior to EV isolation. Urine sample collection in the neonatal population can be challenging as neonates cannot time their voids to easily coordinate a clean-catch sample. Additionally, placing catheters for sterile urine collection introduces the risk of urinary tract infections, which can be particularly dangerous in the preterm population. While Galley et al. were able to isolate and characterize preterm neonatal urine EVs, it is important for future work to develop a standard protocol for urinary EV collection and storage, particularly in the neonatal population.

We were unable to identify studies in which EVs were obtained from neonatal saliva, although their presence in adult saliva [100,101] suggests that EVs are likely to be present in neonatal saliva. The small amount of saliva produced and the inability of neonates to voluntarily provide a sample, pose a potential challenge in sample collection for biomarker discovery. However, a simple bedside suction technique yields between 10 and 50 μL and can be done in extremely premature neonates [12]. Once saliva is collected, commercially available stabilizing solutions can be utilized in this population [102]. Another potential challenge is the effects of hydration status in saliva production, limiting the ability to normalize samples based on volume. To address this, one research group determined that *GAPDH, YWHAZ*, and *HPRT1* are the optimal reference genes for RT-qPCR normalization in neonates as they maintain their stability across various gestational and post-menstrual ages [12,103]. Overall, previous work in neonatal saliva collection and preparation suggests that it can be utilized for EV-miRNAs biomarker discovery.

Many techniques are used to store, process, isolate, and characterize EVs from various biofluids to obtain an accurate yield that will best be used as a clinical biomarker in various disease processes in the neonatal population. Moreover, the disease process may drive which biofluid would be collected for EV isolation and analysis. It is imperative to understand the cellular origin of EVs present in different tissue samples and how enriched pathological tissue-derived EVs in a sample affect their performance as a diagnostic biomarker. A recent study utilizing adult plasma demonstrated that 99.8% of plasma EVs originated from hematopoietic cells and that the remaining 0.2% originated from other tissues [76]. Interestingly, the fraction of EVs derived from liver cells increased in the plasma of patients with hepatocellular carcinoma, indicating that the cell-origin profile of EVs may reflect disease states [76]. Therefore, the cell-origin profile of EVs in a particular sample, by its very nature, has the potential to serve as a diagnostic biomarker. For example, to determine an accurate biomarker for NEC, one may choose to collect and analyze stool, and to determine an accurate biomarker for BPD, one may choose to collect saliva or tracheal aspirates to better obtain and isolate EVs that are more specific to lung pathology. However, to our knowledge, no study has described the cell-origin profile of EVs in neonatal samples.

Additionally, while hUC-EVs may serve as a diagnostic biomarker in specific neonatal diseases, hUC has the major drawback of only providing data from a single time point in a premature neonate’s hospital course. Biomarkers obtained from a single time point with no option to repeat as needed have limited value as diagnostic and prognostic tools in progressive diseases or for monitoring response to treatment.

### 7.2. Validation of EV-miRNAs as Reliable Diagnostic Biomarkers in the Premature Population

As evidenced by the work summarized here, evidence for the use of EV-miRNAs as diagnostic biomarkers is preliminary, with the goal of discovering candidate biomarkers rather than validating the clinical use of such biomarkers. For example, the studies described here have limited sample sizes, with samples obtained from a single center. While most studies reported -*p*-values between experimental and control group EV-miRNAs expression, only two studies reported an area under the curve. Therefore, multi-center clinical studies with larger sample sizes that assess metrics such as sensitivity, specificity, area under the curve, positive and negative protective values, etc., are required to further understand whether the miRNAs discussed here are reliable biomarkers.

As work in preterm EV-miRNAs advances and robust clinical studies are designed, there will be several variables to consider. Determining the right patient population for biomarker discovery and validation will be critical in the clinical application of future work. As evidenced by the work presented here, there is no standardization in what gestational age to include in the preterm population. For example, while some studies for BPD included all neonates <32 weeks gestation, others only included neonates <28 weeks gestation (Table 1). Meanwhile, prematurity is clinically defined as <37 weeks gestation. Since the incidence of NEC and BPD increases as gestational age decreases, will narrowing the gestational age range to those who are extremely premature lead to higher diagnostic accuracy? Additionally, it is unclear how controls will be defined. While some studies here utilized premature neonates without the condition being studied, others used full-term controls (Table 1). Since NEC and BPD are extremely rare pathologies in the full-term neonate, future work should focus on utilizing only preterm neonates as controls.

Furthermore, there is much work to be done exploring EV-miRNAs in preterm HIBD. Most studies not only focus on the full-term population but also do not specifically isolate miRNAs from EVs. While other studies have identified the presence of such HIBD miRNAs within EVs, future studies must be conducted to compare EV-miRNAs between preterm neonates with and without HIBD. This will not only further explore the diagnostic potential of the candidate miRNAs but also identify other candidates EV-miRNAs for validation studies.

### 7.3. Determine the Feasibility of Implementing EV Diagnostic Testing: Testing Population and Role of EV Biomarkers in Clinical Decision Making

Unnecessary laboratory studies increase the risk of negative outcomes and increase the cost of hospitalization, especially when test results lead providers to initiate unnecessary interventions [104,105]. Therefore, future biomarker discovery should focus on a subpopulation of high-risk premature neonates to implement EV-based diagnostic testing. Whether providers should perform EV analysis for NEC or BPD in all premature neonates or only in those who are at high risk of developing such pathology needs to be considered in future studies.

Because the pathogenesis of some diseases in neonates, such as NEC or BPD, is multi-factorial, specific biomarkers may prove to be useful in following disease progression, as well as in direct evaluations and therapeutic options toward a particular pathway of the disease. For example, BPD has been proposed to be the end result of a cascade of events, which may begin with oxygen toxicity, ventilator volume trauma, intrauterine or post-natal infection, and inflammation [75]. Each of these mechanisms contributes to BPD development. Therefore, it is unlikely that the mediators involved in the cascade are identical regardless of the underlying etiology. Using multiple biomarkers from different and distinct biological pathways may differentiate the inciting event and allow pathway-specific therapy directed at the underlying cause of neonatal diseases [106].

The use of EVs and their contents to diagnose conditions in premature neonates may revolutionize laboratory workups and medical interventions in the NICU. However, studies have not specifically analyzed whether the presence of EVs or their contents can aid in clinical decision-making. Researchers are currently designing tools for integration into electronic medical systems that predict NICU length of stay [107], development of BPD [108], NEC vs. spontaneous intestinal perforation [109], sepsis risk [110], and discharge with nasogastric/gastric tube placement [111] in premature neonates. Therefore, future work should analyze whether EVs can provide data points in such tools, contributing to how NICU providers manage care.

## 8. Conclusions

Overall, EVs have a remarkable potential to contribute to our understanding of pathologic processes and aid in biomarker discovery for commonly encountered conditions in premature neonates, including NEC, BPD, and HIBD. There are many opportunities in this field for further characterization of EVs and EV-miRNAs and how they function in such conditions and to address their translation to clinical settings. These include determining how EV heterogeneity affects diagnostic accuracy and whether urine, feces, and saliva can provide a simple, noninvasive method of EV collection.

## Figures and Tables

**Figure 1 ijms-24-02622-f001:**
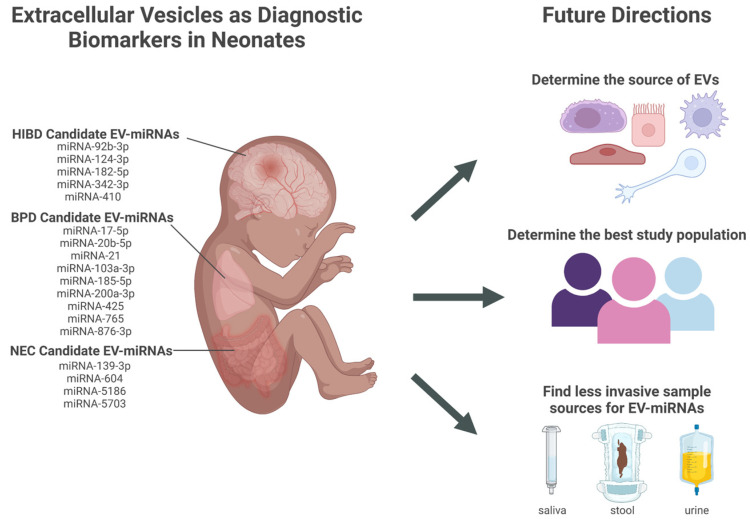
Candidate extracellular vesicles (EVs) and their miRNAs content as diagnostic biomarkers for neonatal conditions and future directions that address challenges in translation to clinical practice. EVs (extracellular vesicles); miRNA (micro-RNA); HIBD (hypoxic-ischemic brain damage); BPD (bronchopulmonary dysplasia); NEC (necrotizing enterocolitis). Created in BioRender.com.

**Table 1 ijms-24-02622-t001:** Summary of EV-miRNAs in NEC, BPD, HIBD.

Condition	Study	Study Population (*n*)	EV Source	EV Isolation (I) & Analysis (A)	EV-miRNA Isolation	miRNA	Statistical Performance
NEC	[64]	**Neonates < 34 weeks GA** NEC (12) Age-matched healthy controls (22)	Urine	I: ExoUrine EV Isolation Kit (System Biosciences) A: NTA, western blot, TEM	ExoRNEasy Midi Kits & Qiagen Qiaquick small RNA Kit (Qiagen Inc.)	139-3p 604 5186 5703	*p* < 0.05 *p* < 0.05 *p* < 0.05 *p* < 0.05
BPD	[66]	**Neonates < 32 weeks GA, DOL 28** BPD (39) Non-BPD controls (34) **Neonatal Mice** Exposed to hyperoxia (4) Exposed to air (controls) (3)	Serum	I: ExoQuick precipitation solution (System Biosciences) A: NTA & ExoScreen, western blot	mirVana miRNA Isolation kit (Ambion Applied Biosystems)	21	*p =* 0.001 AUC = 0.850 *p < 0.01*
	[67]	**Neonates < 32 weeks GA** BPD (12) Non-BPD controls (14)	UC Serum	I: PEG precipitation A: NTA & ExoScreen, western blot, TEM	SeraMir Exosome RNA Purification Kit (System Biosciences)	17-5p 20b-5p 103a-3p 185-5p 200a-3p 765	*p* < 0.05 *p* < 0.05 *p* < 0.05 *p* < 0.05 *p < 0.05* *p < 0.05*
	[68]	**Neonates** 36 weeks PMA with BPD (25) GA-matched, FT controls, intubated for surgery (25) **Neonates < 28 weeks GA** BPD (15) Non-BPD controls (15) **Neonatal Mice** Exposed to hyperoxia (5-7) Exposed to air (controls) (5-7)	TA TA BALF	I: Ultracentrifugation A: NTA	miRCURY RNA Isolation Kit Cell and Plant with miRNA primers (Exiqon)	876-3p	*p = 0.001* *p <* 0.05 *AUC =* 0.917 *p <* 0.05
	[69]	**Neonatal Rats** Exposed to hyperoxia (10) Exposed to air (controls) (10)	Lung Homogenate	I: Total exosome isolation reagent (Thermo Scientific) A: NTA, western blot, TEM	Trizol kits with miR primer (Beijing Dingguo Changsheng Biotechnology Co.)	425	*p <* 0.01
HIBD	[70]	**Neonatal Mice, 24 hours post-surgery** Unilateral carotid ligation + hypoxia (HIBD) (12) Sham surgery + normoxia (controls) (12)	Brain Homogenate	NA	NA	182-5p * 342-3p *	*p <* 0.05 *p <* 0.05
	[71]	**Neonates, FT** Moderate to severe HIBD (7) Healthy control (7)	UC Serum	NA	NA	92b-3p * 342-3p *	*p =* 0.016793 *p =* 0.00059
	[72]	**Neonatal Mice** Hypoxia-preconditioned (3)	Brain homogenate	I: Ultracentrifugation and Sucrose Step Gradient A: NTA, western blot, electron microscopy	RNeasy Lipid Tissue Mini Kit (Qiagen) miRNA-Seq with NEXTFLEX small RNA kit (PerkinElmer)	92b-3p 182-5p 342-3p	NA
	[73]	**Neonates, FT** HIBD (102) Healthy controls (60)	Serum	NA	NA	410 *	*p <* 0.01 AUC = 0.886
	[74]	**Neonatal Rats** Unilateral carotid ligation + hypoxia (HIBD) (12) Sham surgery + normoxia (controls) (12)	Brain homogenate	NA	NA	124-3p *	*p <* 0.05

Necrotizing enterocolitis (NEC); Bronchopulmonary dysplasia (BPD); Hypoxic-ischemic brain damage (HIBD); extracellular vesicle (EV); microRNA (miRNA); Gestational age (GA; day of life (DOL); postmenstrual age (PMA); full-term (FT); tracheal aspirate (TA); bronchoalveolar lavage fluid (BALF); nanoparticle tracking analysis (NTA; transmission electron microscopy (TEM); umbilical cord (UC); not applicable (NA). * miRNAs not specifically derived from EVs but whose presence has been identified in EVs in other studies, as referenced in Section 6.

## Data Availability

No new data were created or analyzed in this study. Data sharing is not applicable to this article.

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
