# Peer review of "Extracellular Vesicle-microRNAs as Diagnostic Biomarkers in Preterm Neonates"

_ijms, 2023, doi:10.3390/ijms24032622_

Round 1

Reviewer 1 Report

The manuscript of Emily A. Schiller et al. is a review demonstrating the perspectives of the use of extracellular vesicles/EVs (particularly microRNAs and EVs cargo content) for as disease biomarkers  in preterm neonates. It is and interesting approach, since EVs have attracted much interest in the past decade due to their potential utility as circulating biomarkers for cancer. However,  the current form of the manuscript is not suitable for publication in high impact factor Journal as IJMS and further significant improvements and revisions are necessary.

 General Comment: In my opinion, this review manuscript prepared is somewhat cursory, abbreviated and does not address important aspects of the use of EVs as disease biomarkers. Accelerating the translation of EV-based biomarkers from the bench to the bedside will require researchers to utilize robust methodologies to overcome these hurdles, thus allowing for the establishment of a gold standard of EV isolation and analysis and to identify clinically applicable EV based biomarkers for diagnosis and monitoring of diseases. There are now a large number of published review papers discussing the possibility of the use of EVs phenotyping, their content composition or exosomal messenger RNA (mRNA), small endogenous exosomal noncoding RNAs (miRNAs), long noncoding RNAs (lncRNAs) as biomarkers of given diseases; however, the main problems are, among others: standardization of measurement protocols performed, unequal study group, too small sample size, etc. The authors have not analyzed or even tried to discuss these problems, which makes the scientific level and soundness of the work quite low, in my opinion.

The International Society of Extracellular Vesicles (ISEV) several years ago prepared the list of minimal information for studies of extracellular vesicles (MISEV2018/ https://doi.org/10.1080/20013078.2018.1535750), covering extracellular vesicle (EV) separation/isolation, characterization, and functional studies.  The purpose of this recommendation was to “sensitize researchers, as well as journal editors and reviewers, to experimental and reporting requirements specific to the EV field (…) when making strong conclusions on the involvement of EVs, or specific populations of EVs (exosomes in particular), in any physiological or pathological situation, or when proposing EVs or their molecular cargo as biological markers”. The ISEV directly indicating that major conclusions in some articles are not sufficiently supported by the experiments performed or the information reported.

In my opinion also in case of the submitted manuscript, the manner of results description, lack of important information about the EVs isolations, purification procedures, used protocols and methodology significantly limit the scientific level of this manuscript.

Detailed comments:

1) Only a well-defined, statistically powered study, with positive and negative controls, and using consistent samples collection, processing and storage using reproducible EVs isolation and analysis protocols and data analysis pipelines will produce clinically useful biomarkers. It is also important to report biomarkers with the same rigorous standards with which they are developed.

In the submitted manuscript, the results related to the cited researches were briefly described without direct information about very important issues such as  used protocols and methodology that should be reported. In my opinion, the authors should expand the scope of a manuscript by including the detailed overview of the common methods/procedures of EVs isolation, purification (see ISEV recommendation: https://doi.org/10.3402/jev.v4.27031) and EVs phenotyping strategies related to diseases or disorders discussed in the manuscript.

2) Moreover, they should perform a complex discussion and comparison of the all analyzed results, used methodology, protocols according to the ISEV recommendations  ( see examples below):

https://doi.org/10.1080/20013078.2018.1535750

https://doi.org/10.3402/jev.v2i0.20360

https://doi.org/10.3402/jev.v4.27031

https://doi.org/10.3402/jev.v3.26913

Only a fair comparison of the proposed method according to generally accepted standards will ensure that this manuscript meets the requirements of a review paper and will increase its scientific potential, without introducing unnecessary confusion to the audience.

3) The authors describe the results of performed using arbitrarily chosen references related to the scope of the manuscript, but there is no information about the number of all participants whose biological samples were taken and used for research purposes. Without such information, it is hard to objectively determine and verify whether the results obtained are representative or sufficiently scientifically supported.

4) The complexity and heterogeneity of biological fluids becomes essential to minimize pre-isolation and pre-analytical variables through the standardization of sample collection, storage, and handling.  However, the authors in this case did not analyze these issues. They should present and discuss some recommendations and measures to minimize artifacts for downstream isolation and analysis.

5) Describing the results of researches, the authors are not providing any patients characteristics, inclusion/exclusion criteria, etc. This information should be provided.

6) The authors indicated and investigated only miRNAs as biomarkers in the submitted manuscript, but even with a limited scope, authors did not decide to introduce any comparison of common measurements strategies in miRNA detection, detection sensitivity, or detection (at nano-/femto-/attomolar concentrations ?). Such a consideration should be included in this manuscript. The authors should add an overview and comparison of the characterization methods in the references EVs, which were applied in cited or can be used in the diseases/disorders discussed in the manuscript.

7) I wonder why the authors mention miRNA-21 among potential biomarkers, since it was already reported (https://doi.org/10.1186/s40364-021-00272-1 ) that ‘miR-21 cannot be considered a specific biomarker for any disease if it is a biomarker of many diseases. While its levels may genuinely vary across bodily fluids in disease states, these variations have no specificity”. Therefore, I wonder why the authors decided to consider such nonspecific miRNA as a biomarker of diseases.

8) There is no information in the research manuscript as to whether the performed from cited references were standardized according to the international requirements recommended by ISEV. The  growing interest in EV makes the technical standardization extremely important,  because many methodologies can be used to isolate and analyze EV. Different isolation procedures, a variety of techniques used to purify RNA,  can significantly affect extracellular RNA sequencing and profiling, making the results unclear. Therefore, authors are asked to provide detailed information (which may be in a form of the table) describing the comparison of the discussed diseases/neonates disorders, related miRNA biomarkers, volume of biofluid method/procedures of isolation of EVs/purification of a subset of EVs, examined sample size (sample size refers to the number of all participants whose biological samples were taken and used for investigation purposes), miRNA detection principles, sensitivity, LODs, etc.

9)  What about the analytical specificity of miRNas indicated as possible biomarkers? This issue is not discussed in the manuscript.

10) The authors present some perspectives and concepts of the use of miRNA from EVs in Sections 7 and 8. In Section 7.1, they tried to described the best sample source for discovering diagnostic biomarkers from neonatal EVs. However, this discussion is very cursory, and this aspect is briefly described. The authors did not analyze problems, limitations, advantages/disadvantages of the selection of EV source. For example, there are a wide range of techniques that can be used for isolating EVs from biofluids, which have been reviewed extensively. A key area of difference between recent studies is the biofluid used to isolate EVs. Although EVs have been isolated from almost all bodily fluids, the main sources commonly used include urine and blood (both serum and plasma). One of the most abundant sources of EVs is blood, however, the most challenging aspects of the isolation of EVs from blood is that it contains lipoproteins and chylomicrons, which overlap in size and density with EVs and cannot be completely removed by conventional isolation methods. Moreover, the amount, purity, and heterogeneity of EVs from blood are influenced by sample collection, handling, storage conditions, stability, anticoagulants, volume of blood collection, time of blood collection, and the age, sex, disease state, and fed/fast status of the animal/patient. Urine is very accessible, with collection simple, pain-free, and noninvasive. However, urine composition shows high variation, is easily influenced by dietary, medicinal, and diurnal changes, and is particularly susceptible to collection error by patients.

11) In the Introduction section the authors did not describe any recent advances in the measurement techniques focused on the EV phenotyping including SERS, digital holotomography, nanoparticle tracking analysis (NTA), AFM, quantitative polymerase chain reaction etc. (see some examples below)

https://doi.org/10.1002/adfm.202010296

https://doi.org/10.1002/mnfr.201500222

https://doi.org/10.1039/D0NR07349K

https://doi.org/10.1016/j.jconrel.2018.08.035

https://doi.org/10.1016/j.ajpath.2021.08.005

https://doi.org/10.1038/s41467-020-14344-7

DOI 10.1088/1361-6528/aaab06

https://doi.org/10.1007/s11051-014-2583-z

The authors did not also describe modern methods of miRNA detection based e.g. on interferometric biosensors, SPR biosensors, localized SPR biosensors, UV-Vis method, electrophoresis-based techniques, the RiboGreen assay, quantitative reverse transcription (qRT)-PCR-based assays etc. (see examples below):

https://doi.org/10.1016/j.bios.2015.04.052,

https://doi.org/10.1016/j.bios.2021.113613,

https://doi.org/10.1038/s41467-018-07947-8,

https://doi.org/10.1016/j.bios.2020.112599,

https://doi.org/10.1021/acsomega.1c06479,

https://doi.org/10.1021/nl503220s,

https://doi.org/10.1021/acsnano.5b04527,

https://doi.org/10.1021/acsnano.5b04527,

Author Response

THANK YOU for your consideration and your very helpful comments. They improved our manuscript significantly. Point by point response is in the attached docuemnt. 

Reviewer 2 Report

 Before of presenting abbrivation, words should be introduced full.

BPD? 

HIBD?

There are no citation for some sentences. For instance: 

Lines 190 to 194  

243-9

250-2

130-135

233-5

243-9

257-259

What is the sensitivity and specificity, accuracy of  Extracellular Vesicles in detection of NEC,  HIBD,  BPD in neonates?

Lines 85-9 must be omitted. This study is about EVs not  microRNAs

Line 89: mRNAs???

Lines 107-120, 137-147, 196-205 should be omitted. All these sentences are repetition of the literature review. 

Author Response

THANK YOU for your consideration and your very helpful comments. They improved our manuscript significantly. Point by point response is in the attached document. 

Reviewer 3 Report

See attached file

Author Response

(The authors gave the same response as above.)

Round 2

Reviewer 1 Report

I would like to thank the authors for the clarifications sent and corrections made to the manuscript.

Especially section 7.1 and Table 1 have enriched this manuscript and raised its scientific quality. The authors have done a very good job. When using extracellular vesicles, the way the sample is obtained and the procedures for their preparation play a crucial role, but these are quite often overlooked, making it impossible to standardize the results obtained.

In my opinion, the manuscript in its present form, i.e. after the corrections made, is suitable for publication in the International Journal of Molecular Sciences.

Author Response

We would one more time to thank this reviewer for taking the time to provide us with the very constructive feedback. These suggestions have increased the scientific quality of our paper significantly.  

We also corrected a few typos we noticed. The revised manuscript includes all corrections. 

Reviewer 2 Report

Dear authors, many thanks for the revision,

Please cite the reference for lines 56-67 of page 2.

Author Response

We would like one more time to thank this reviewer for taking the time to provide us with very constructive feedback. These suggestions have increased the scientific quality of our paper significantly.  

As requested by the reviewer, we added the requested references (7,11,12 in line 66) after the sentence. The revised manuscript includes all corrections.